# Does Engagement Build Empathy for Shared Water Resources? Results from the Use of the Interpersonal Reactivity Index during a Mobile Water Allocation Experimental Decision Laboratory

**Lori Bradford** [1] , **Kwok P. Chun** [2] , **Rupal Bonli** [3] **and Graham Strickert** [1,*]

[1]  School of Environment and Sustainability, University of Saskatchewan, Saskatoon, SK S7N 5A2, Canada;
    lori.bradford@usask.ca
[2]  Department of Geography, Hong Kong Baptist University, Baptist University Rd, Kowloon Tong,
    Hong Kong 999077; kpchun@hkbu.edu.hk
[3]  Saskatoon Health Region, Saskatoon City Hospital, 701 Queen Street, Saskatoon, SK S7K 0M7, Canada;
    rupalbonli@shaw.ca
*  Correspondence: Graham.Strickert@usask.ca; Tel.: +1-306-966-2403

**Abstract:** Currently, there are no tools that measure improvements in levels of empathy among diverse water stakeholders participating in transboundary decision-making. In this study, we used an existing empathy scale from clinical psychology during an Experimental Decision Laboratory (EDL) where participants allocated water across a transboundary basin during minor and major drought conditions. We measured changes in empathy using a pre-test/post-test design and triangulated quantitative results with open-ended survey questions. Results were counter-intuitive. For most participants, levels of the four components of empathy decreased after participating in the EDL; however, significant demographically-driven differences emerged. Qualitative results confounded the problem through the capture of participant perceptions of increased overall empathy and perspective taking specifically. Implications for methodological tool development, as well as practice for water managers and researchers are discussed. Water empathy is a particularly sensitive construct that requires specialized intervention and measurement.

**Keywords:** water security; empathy; interpersonal reactivity index; experimental decision laboratory; transboundary water management

---

## 1. Introduction

The notion of bringing the right number and representation of diverse stakeholders with the right frequency to the decision table for shared resources like water is understood as a key practice for increasing resource security [1,2]. Participatory approach frameworks, toolkits, and reviews abound for a variety of resource security issues [3–5]. The effectiveness of participatory approaches for enhancing security of resources has been investigated with key performance indicators typically chosen by hierarchical agents [6,7]. The indicators have included species biodiversity and/or population declines, nutrient balances, emission and pollutant levels, resource use, energy use, and other biophysical measures [8,9]. The effectiveness has been evaluated using absolute data, trend data, and normalized data and effectiveness studies have started to include human dimensions such as participant learning [10,11]. An area lacking in evaluation of participatory resource security performance has been the objective evaluation of whether participants experience an increased level of awareness and

sensitivity to each other's plights. It is hypothesized that such an increase would contribute to more equitable sharing of resources and thus security [12].

In this paper, we describe an approach for measuring 'water empathy' among stakeholders participating in natural resources decision scenarios which involved allocating water across three regions, and four broad sectors of society within each region in a transboundary river basin. While the drivers of water security in the study basin are examined elsewhere [13], this paper specifically examines the use of an empathy scale from clinical psychology to better understand whether the experimental decision laboratory could invoke changes in people's ability to share and understand the thinking and feelings of other stakeholders in water allocation decisions.

*Empathy, Water Empathy, and Its Measurement*

There are many different definitions of empathy and a long history of debate over its constructs [14–16]. In clinical and developmental psychology, empathy is defined as the act of putting oneself in the other's place, a psychological stance toward another with intentionality, and is differentiated from confronting, advising, correcting or teaching [17–20]. Empathy develops through a two-part process: (1) establishing and maintaining affective resonance with another, and (2) learning to change this resonance into shared language and mutually beneficial action [21]. Psychological scales measuring empathy emerged in the 1950s and evolved to include affective components such as emotional distress, and cognitive components such as perspective taking [22–28].

Changes in participant levels of empathy have been studied indirectly in various resource management contexts. Explorations of social capital and self-interest in farming, forest management, fisheries, and community-based collaborative resource management groups indicate three key findings. First, participants rely on past experience and habits more often than previously recognized for decisions with downstream impacts [29]. Second, social capital is enhanced among groups and agencies but not between individuals [2,30,31]. Third, there is little emphasis on listening to all participant needs at the outset of resource management decision making, and instead value struggles are dealt with post hoc [32,33]. These three problems contribute to poor management decision outcomes such as resource conflicts, and lead to the conclusion that policies need to invoke mutual understanding among participatory resource management groups [34–36].

In this study, the term water empathy is coined as the ability to understand and be responsive to the water needs of another group or sector using a shared water resource. Acts of water empathy have been noted in the past with physical actions, such as senior license owners in a heavily allocated Canadian river system transferring allocations to help other sectors and, social learning, such as collaborative information sharing in Australia's Lower Burdekin for better natural resource management and decreased impacts to the Great Barrier Reef [37,38]. In the context of water management, few studies to date have explored whether current participatory research approaches examining policies for transboundary water contributed to creating empathy among water managers and stakeholders and thus better resource management performance. Anecdotal evidence suggests that it does [1]. To this end, this study was undertaken to test a psychological empathy scale as a starting point towards developing a tool for measuring changes in water empathy among water managers, stakeholders and right's holders across a transboundary river basin during engagement activities.

## 2. Materials and Methods

As a part of a larger research program examining the human dimensions of water security, water allocation preferences, and management practices across the Saskatchewan River Basin in the Prairie Provinces of Canada, a mobile experimental decision laboratory (EDL) was developed. The EDL was a computer-based simulation of a transboundary river basin that was undergoing a mild or severe drought and was based on conditions in the history of the Saskatchewan River Basin, Canada. After ethics approval was received (University of Saskatchewan, BEH-13-386), participants who were involved in water use, management, energy sectors, Indigenous communities, and water research were

contacted using phone and/or email invitations to take part in one of the EDL workshops. A letter and consent form describing the research and its aims was distributed prior to the EDL occurring. During the EDL itself, participants went through several rounds of having to allocate water to different sectors under different policy constraints and communication boundaries. These boundaries (i.e., separation of participants for certain rounds, timed communication opportunities) were kept consistent. Instructions were scripted so that each occurrence of the EDL followed the established protocol. The EDL was a mobile unit; that is, personnel and computers/other equipment were transported to different cities within the basin over a four-month period. Participants used the exact same equipment in each EDL event. Five different EDL events took place between January and April 2015 in Canmore, Alberta; Medicine Hat, Alberta; Northern Village of Cumberland House, Saskatchewan; and two in Saskatoon, Saskatchewan, Canada. The EDL protocol and results have been reported elsewhere [39]; however, out of interest in evaluating empathy among water managers, pre- and post-test of levels of empathy were included in the EDL sessions. This paper reports on those results. The researchers collaborated with a clinical psychologist to select an appropriate scale, pilot test the scale, and evaluate the results.

*2.1. The Davis (1980) Interpersonal Reactivity Index*

The Interpersonal Reactivity Index (IRI) measures four components of empathy, three of which are of an affective (emotional) nature and a fourth component that is more cognitively-based [40]. The three affective components include fantasy (the tendency to identify strongly with fictitious characters), empathic concern (the tendency to experience feelings of compassion and concern for others undergoing negative experiences), and personal distress (the tendency to experience feelings of discomfort and anxiety when witnessing the negative experiences of others) [26]. The more cognitively-based 'perspective-taking' component reflected a tendency or ability of the respondent to adopt the perspective, or point of view, of other people. The IRI consists of twenty-eight items; seven different items for each of the four components scored on a five-point modified scale asking participants to what extent each item describes them. The highest possible mean score for each individual component was 5, and the lowest was 1. The items were presented in random order to participants. In this study, at least two items per component were phrased in the negative. The scale reliability and validity were reported as high with internal reliabilities ranging from 0.70 (men) to 0.78 (women) and test–retest reliabilities ranging from 0.61 to 0.79 (men) and 0.62 to 0.81 (women) over 60–75 days [26]. The scale used in this study is provided in Appendix A.

*2.2. Protocol and Measurements*

Participants were administered the scale on personal computing devices (i.e., iPad) as they awaited the start of the EDL and again immediately after concluding the EDL. Data entry was verified for completeness and accuracy by two researchers by proofreading every third entry. A follow-up email was distributed one week after the EDL with a survey of open-ended questions about perceived changes in empathy (Appendix B). Ages were computed into three ranges as it is indicated that some empathic components decrease with age [26]. All negatively phrased items were transformed to positive ones in the database (i.e., a 2 out of 5 on a negative item became a 3 out of 5 on the positive). Group scores of the four components of empathy were computed by averaging the mean scores for the seven items for each component of empathy from each individual. A further summary mean score from the means of the three components of affective empathy was computed [40]. Results were subject to Kruskal–Wallis and Wilcoxon Signed Rank (with effect sizes converted from eta-squared to Cohen's d where possible) among males and females, riparian location, water stewards' employment roles, and age brackets.

## 3. Results

### 3.1. Participants

Thirty-seven participants were recruited (Table 1). All participants were employed in the water management sector and had previously been involved in conferences and workshops and completed surveys as a part of the larger research program. Participants were from various locations across the Saskatchewan River Basin including Canmore and Medicine Hat among others Upstream (N = 6), Saskatoon and Lethbridge in Midstream (N = 27) and Nipawin and Cumberland House Downstream (N = 4). Although the sampling goal was not to gather enough participants for ease of generalizability across the population, the participants approximately represented the demographic particulars of resource managers in the basin and the researchers were confident that no water allocation perspectives were missed given their previous research (see [39,41]).

**Table 1.** Sample characteristics.

| Location | Gender | Mean Age (mean 36.4) | First Nation/Métis: Non-Indigenous | Water Mgt. Role* |
|---|---|---|---|---|
| Upstream (N = 6) | 4 Female 2 Male | 43.3 (range 24–73) | 1:5 | 1 URWU, 2 Private 1 RAWU, 2 Govt. |
| Midstream (N = 27) | 12 Female 15 Male | 32.4 (range 22–71) | 1:26 | 6 URWU, 5 Private 7 RAWU, 4 Govt. 5 Research |
| Downstream (N = 4) | 2 Female 2 Male | 34.0 (range 21–49) | 2:2 | 1 Govt., 2 Private 1 Research |
| Total (N = 37) | 18 Female 19 Male | 36.6 (range 21–71) | 4:33 | 7 URWU, 9 Private 8 RAWU, 7 Govt. 6 Research |

\* Water Management Roles included Urban Residential Water Users (URWU), Rural Agricultural Water Users (RAWU), Private Users (Water Provider, Private Sector, Non-Governmental Organization, Other), Government Agency (Federal, Provincial, Municipal, Tribal), or Researcher.

During the EDL, participants used laptop computers to make water allocation decisions under three different policy scenarios, through two different levels of water stress and under two different communication rules (can or cannot communicate with other users). There were six separate rounds of allocation [40]. Participants were told to imagine themselves acting as the water managers in the fictional basin and make decisions as 'official' allocators for a randomly selected region in that basin. They had to act under the different policies (for example, The Master Agreement on Apportionment) and communication rules (for example, cannot talk to each other at all, or can talk with other participants for a total of 5 minutes) but had leeway to allocate the water to different sectors (agriculture, industry, municipal users, and environmental services). Communication rules and workshop logistics meant that, for the first four scenarios, EDL participants could not communicate with other participants, while, for the last two, they could for a limited amount of time. The purpose of this paper is to report solely on the effects of participating in the EDL on water empathy, although it is recognized that the protocol for the EDL may have influenced other social psychological factors that were not measured with the given empathy scale, nor were part of the original experimental design. We also did not ask about each participant's prior experience with the others they were undergoing the EDL with at the time, or with empathy scales in general.

### 3.2. Empathy Component Pre and Post-EDL Scores

Overall, data were non-normally distributed, but had the same significant negatively skewed shape (Pre-test skew = −3.304, Post-test skew = −3.364, standard error both 0.388, $p < 0.05$). The total empathy across the combined IRI test scores decreased overall but non-significantly (Pre-test mean 2.861 to Post-test mean 2.846, W = −0.173, $p = 0.863$). Though the perspective taking scores (N = 37) increased

significantly driven in small part by gender (d = 0.210), the other three components overrode that increase for total empathy (Table 2). The mean fantasy scores for the pre- and post-test (2.790 to 2.636) decreased significantly (W = −1.543, *p* = 0.041) but was not driven by gender or age. The empathic concern component decreased significantly, driven by men (W = −2.079, *p* = 0.036, d = 0.130), and the personal distress component (W = −0.371, *p* = 0.710) did not change significantly overall. However, differences between age groups were apparent with younger participants developing significantly increased mean ranks for personal distress (i.e., experienced increased self-focus discomfort over another's plight) versus older participants (Post H = 6.140, *p* = 0.046*, d = 0.122). The summative affective component, however, had a significant decrease (W = −2.167, *p* = 0.011) driven by men (effect size d = 0.19). The significant findings were driven in small amount by gender; women scored themselves higher to begin with on all empathy components (mean scores of 3.018 compared to men 2.712); most strongly for perspective taking; and women had markedly less differential between pre- and post-test scores. Scores for perspective taking increased significantly for women (H = 2.718, *p* = 0.049, d = 0.21) while the decrease in empathic concern was driven by the scores of the male participants (H = 6.909, *p* = 0.009, d = 0.14) which also influenced the affective summary score.

**Table 2.** Pre- and Post-Experimental Decision Laboratory empathy survey results.

| Test Items (N1 = N2 = 37) | Perspective Taking (ΔM, W, p, d when indicated) | Fantasy (ΔM, W, p, d) | Empathic Concern (ΔM, W, p, d) | Personal Distress (ΔM, W, p, d) | Affective Components (ΔM, W, p, d) |
|---|---|---|---|---|---|
| Overall (N = 37) Wilcoxon signed rank | 1.676, W = −3.195, *p* = 0.001* | −0.154, W = −1.543, *p* = 0.041* | −0.108 W = −2.079 *p* = 0.036* | −0.039 W = −0.371 *p* = 0.564 | −0.099 W = −2.167 *p* = 0.011* |
| **Gender-related Analyses (Kruskal–Wallis H test, p, r (Driven by men/women))** | | | | | |
| Pre-test (N = 37) | H = 0.393 *p* = 0.331 | H = 1.881 *p* = 0.170 | H = 4.912 *p* = 0.027* d = 0.32 (men) | H = 2.417 *p* = 0.120 | H = 4.543 *p* = 0.033* d = 0.19 (men) |
| Post-test (N = 37) | H = 2.718 *p* = 0.049* d = 0.21 (women) | H = 1.716 *p* = 0.190 | H = 6.909 *p* = 0.009* d = 0.14 (men) | H = 0.628 *p* = 0.428 | H = 3.498 *p* = 0.061 |
| **Age-related analyses (Wilcoxon Mean Rank Scores)** | | | | | |
| **Age 21–34** (N = 20) | | | | | |
| Pre-test Mean Rank Post-test Mean Rank | 17.15 15.38 | 22.03 21.70 | 17.78 18.43 | 19.68 22.38 | 20.45 21.85 |
| **Age 34–50** (N = 11) | | | | | |
| Pre-test Mean Rank Post-test Mean Rank | 20.50 24.55 | 14.36 14.50 | 20.41 19.91 | 21.00 17.68 | 17.77 15.23 |
| **Age 50+** (N = 6) | | | | | |
| Pre-test Mean Rank Post-test Mean Rank | 22.42 20.92 | 17.42 18.25 | 20.50 19.25 | 13.08 10.17 | 16.42 16.42 |
| Age group: Wilcoxon test, p, d | Pre-test W = 1.409 *p* = 0.494 Post W = 5.369, *p* = 0.068 | Pre-test W = 3.731 *p* = 0.155 Post W = 3.190 *p* = 0.203 | Pre-test W = 0.564 *p* = 0.754 Post W = 0.139 *p* = 0.933 | Pre-test W = 2.260 *p* = 0.323 Post W = 6.140 *p* = 0.046* d = 0.122 | Pre-test W = 0.845 *p* = 0.655 Post W = 3.069 *p* = 0.216 |

* Significant at α = 0.05 level, d = Cohen's d effect size.

The change in the components of empathy were also driven by age-related effects (Table 2). Older participants (aged 50+) exhibited less affective empathy after the EDL, especially for personal distress when compared with the youngest participant group for which the EDL caused a small increase in feelings of personal distress (H = 6.140, *p* = 0.046, d = 0.122). Additional analyses were sought by participant riparian position; however, the small sample of downstream participants meant no significant differences were calculable across upstream, midstream, and downstream groups. Participants also hailed from different water management employment sectors in the Saskatchewan

River Basin. Significant changes in sector-by-sector empathy scores across the four components and the summative affective component are listed in Table 3.

**Table 3.** Sector-based changes in empathy.

| Sector of Water Employment (N) (Women, Men) | Perspective Taking (ΔM, Pre-Test: Post-Test Rank) | Fantasy (ΔM, Pre-Test: Post-Test Rank) | Empathic Concern (ΔM, Pre-Test: Post-Test Rank) | Personal Distress (ΔM, Pre-Test: Post-Test Rank) | Affective (ΔM, Pre-Test: Post-Test Rank) |
|---|---|---|---|---|---|
| Rural Agricultural Water Users N = 8 1W, 7M | −0.125 15.63:11.44 | −0.393 16.13:13.94 | −0.250 10.38:9.50 | −0.054 15.63:15.31 | −0.232 12.81:11.19 |
| Government Agency Representatives N = 7 3W, 4M | 0.449 19.07:23.21 | 0.073 22.29:23.64 | 0.020 20.93:23.00 | −0.143 22.21:20.64 | −0.041 22.71:24.29 |
| Urban Residential Water Users N = 7 4W, 3M | 0.225 16.43:15.07 | 0.041 24.21:26.14 | −0.143 27.64:25.71 | −0.082 26.57:27.57 | −0.061 28.21:29.79 |
| Researchers N = 6; 5W, 1M | 0.476 25.75:27.75 | −0.247 18.83:17.50 | −0.024 16.92:18.75 | 0.095 21.42:23.50 | −0.056 20.33:20.25 |
| Private Users N = 9 4W, 5M | 0.254 19.44:19.67 | −0.159 15.06:15.33 | −0.111 19.83:19.28 | 0.043 12.00:11.33 | −0.085 13.56:12.61 |
| **Wilcoxon signed rank test, *p*, d** | W = 9.936 *p* = 0.042* d = 0.13 for Govt. Reps and d = 0.09 Researchers | W = 7.270 *p* = 0.020* d = 0.10 for RAWU | W = 9.935 *p* = 0.038* d = 0.08 for RAWU | W = 11.097 *p* = 0.025* d = 0.11 for Govt. Reps | W = 16.027 *p* = 0.003* d = 0.09 for Govt. Reps |

* Significant at $\alpha$ = 0.05 level, ΔM = Post-test mean—Pre-test mean, d = Cohen effect size.

Sector-based results indicate that rural agricultural water users exhibited decreases in all components of empathy after participating in the EDL, contributing small significant effects to the decreases in fantasy (W = 7.270, $p$ = 0.020, d = 0.10) and empathic concern (W = 9.935, $p$ = 0.038, d = 0.08). Only the rural agricultural water users exhibited decreased perspective taking. Government agency representatives exhibited a small but statistically significant increase in perspective taking, and increases in fantasy and empathic concern, but an overall decrease in total affective components. The participants classified as "Private Users," and Researchers both exhibited significant change in personal distress and the Researchers had significant decreases in the fantasy components (Table 2).

*3.3. Qualitative Data*

The follow-up survey results revealed differences in the perceived effect of participating in the EDL on empathy. Over 60% (23/37) believed participating in the EDL provoked a change in the way they thought about water management across a basin. Additionally, 14 participants provided in-depth qualitative responses about how their thinking changed as a result of the EDL experience. These were thematically-coded using a grounded approach into three drivers for their allocation decision-making. First, participants expressed that the exposure to different water user needs and viewpoints challenged their former values about allocating water:

> The experiment and the conference both allowed me to talk with other stakeholders and further learn about their concerns and sometimes their solutions. I still hold onto my values, but can see the value in trying to strike a balance. P5—Male Upstream Govt. representative

> After being a water decision maker, and having to decide for other sectors, after, I felt more sympathetic to irrigation and industry. I know they have water needs and that environment, should not necessarily have priority over them. I struggled because of my previous beliefs. I thought if I change my opinion now, there is one less person advocating for environment and

the balance will be even more out of wack (sic) P24—Female Midstream Urban Residential Water User

Secondly, participants reported that new knowledge of industry and economic values affected their water allocation decisions:

I changed how I thought about water allocation as an economic foundation. I often do not think about how much water is used by various industries. I also started thinking about how it affected the economy of those downstream. P8—Male Midstream Private Industry

I also realize I may have to rethink my initial bias toward environmental demands and against economic. Both are important, but I think we have to focus on demand that maximize our economic output while sacrificing the least amount of environmental demands as possible. P35—Male Midstream Researcher

When I think of Industry or agriculture, I think of greedy sectors, with no boundaries, that they will take and take and take, at all costs. As a water decision maker, seeing the sector's needs, and that there were finite limits to satisfy, changed that perspective of greed. P25—Female Upstream Govt. representative

Third, participants relayed that they learned more about their upstream and downstream neighbors and thereby provoked self-reflection:

I realized how challenging it is to decide how to distribute water under dry circumstances. I also didn't ever really understand how much upstream users affect downstream users. P34—Male Midstream Rural Agricultural Water User

The quote "when the watering hole gets smaller, the animals look at each other differently" ran through my head throughout the experiment and workshop. This will happen to us in times of water shortage, but hopefully we will talk with each other differently rather than look at each other differently. P5—Female Midstream Govt. representative

'Realized that listening to other people needs are really important, there is always enough to make it go around. The importance of having some safety measures and (in) place, as long as it does (not) conflict the rights and freedom of the people and its made to benefit everyone. P14—Male Upstream Rural Agricultural Water User

These three drivers were consistently reported across the open-ended survey questions, but a fourth point was raised by a few participants:

Speaking briefly with the other two showed that we did have differences. P1—Male Midstream Private Industry

There were some different point of views brought up during the third [shared risk] discussion with that provided me with different viewpoints on things I hadn't thought of during the first run through. P3—Male Midstream Researcher

In a collaborative environment talking helped to make allocations more equitable but in political arena rules are probably needed. P16—Male Upstream Private Industry

I think striving for equal allocation is the best first approach and if that is not feasible then all users have to start communicating and consider the demands of each region and which demands have the most benefits for the majority of users and factor that into further allocation decisions. P20—Female Downstream Researcher

These statements refer to the part of the EDL where participants were instructed that they could communicate with upstream and downstream participants before making their allocation decisions. The statements reflect the positive interaction and shared learning that occurred during that segment of the EDL.

The results appear counter-intuitive; that is, the research team believed, and the qualitative reflections from the participants support the assertion that participating in the EDL and having the opportunity to interact with different water stakeholders from across the basin enhanced the understanding of water needs. They also believed that the experience improved allocation decisions in different regions and the understanding of values for difference employment roles. Thus, there is the belief that the EDL enhanced water empathy to some degree. The IRI scale results, however, indicated that the summative affective components of empathy decreased as a result of the EDL experience. In addition, perspective taking was enhanced for women, but not men and rural agricultural water users, and urban water users and government representatives had enhanced fantasy scores. A factor contributing to these results could be that the qualitative data was collected a week later, thereby allowing for some reflection to occur, while the IRI was completed immediately after the EDL and on site, with the possible interference of participant fatigue and experience of the IRI tool. The discussion will shed some light on the implications of these results.

## 4. Discussion

### 4.1. Affective Components, Personal Distress and the EDL

It has been put forward that increasing one's sensitivity to the challenges of others would assist in navigating difficult conversations about sharing resources [12]. Affective empathy, or emotional empathy, is likened to emotional contagion and inducing sensitivity to others. In psychological contexts, affective empathy is a key ingredient in successful relationships [42]. The EDL hoped to contribute to this relationship-building between diverse participant groups in the Saskatchewan River Basin through increasing understanding of the emotional difficulties faced by other resource users due to allocation decisions. The decrease to the affective components of empathy across the participant groups in this study is surprising, but has several potential drivers. First, the personal distress component of empathy as a measurable component itself could be flawed. That is, other studies have concluded that personal distress is a measure that reflects emotional maturity and cognitive development as one transitions from childhood to adulthood, and is actually more representative of sympathy (identifying with another person, but still focusing on the self and protection of the self) [43]. Though the sample was relatively young (mean age 36.6), there is evidence of increased sympathy in the qualitative results, for example, in feeling more sympathetic to another sector, or changing one's initial bias. On the other hand, personal distress is described as an individual's fear, feelings of apprehension and discomfort at witnessing the negative experiences of others [40]. When people age or grow in experience in a particular context, this component decreases due to gains in emotional control, and fewer fears of what a person has witnessed happening to them. Instead, there is a shift to feeling sympathy for victims and acting in pro-social ways [44,45]. The EDL experience was designed around creating a safe space for allocating a resource and perhaps lacked evidence of the experience of victims of poor allocation decisions. This would partly explain why only some participants (i.e., younger, and those who tend to work in offices, not on the land) experienced a shift in personal distress; that is, they learned a little of what the experience might be like when they otherwise would have no experience with the consequences of allocation decisions. The rural agricultural water users, in contrast, would likely have more lived experience of water shortages and floods over their careers, and thus feel a reinforcement of beliefs of the consequences of poor decision-making for them personally, as well as a desire for others to know that they experience water stress too. Evidence to this can be found in the qualitative results, for example, changing perspectives on the 'greed' of agricultural users by other groups, and the agricultural user's comment about protecting everyone's rights and freedoms. In summary, the

decreases in reported levels of personal distress could be a cohort effect as reported in other studies using the IRI [46,47], and the decrease in affective empathy may be driven by the diversity of the participants. Researchers studying empathy in resource management challenges should be aware of the potential for polarization of emotions and reinforcement of personal distress when different sectors come together in engagement activities.

### 4.2. Fantasy and the EDL

The fantasy component of empathy had an overall significant decrease in this study, driven mostly by the rural agricultural water users and researchers. Recall that the fantasy scale items catch a movement away from identifying with fictitious stories or characters in a given situation [26,48]. It may be that the EDL experience adds a bit of practicality to an otherwise unfamiliar situation for rural agriculturalists used to living on large farms more upstream and unaware of downstream struggles, and for researchers who are typically far-removed from the technical or on-the-ground tacit experience of managing water allocations. It is also arguable whether or not the fantasy component reflects an affective or cognitive responses. There is debate in the fields of psychology and neurosciences providing mixed conceptualizations of this particular empathy construct [45,49,50]. Being able to differentiate realistic and unrealistic scenarios resulting from water allocation decisions may influence a participant's ability to empathize with others, and, in this case, could reflect cognitive processing rather than emotional relationality among participants. In this study, some of the qualitative results support this assertion in that they demonstrate participants' processing of the experience beyond an emotional level, for example, when P5 states that they are holding onto their values, but can see the benefit of 'striking a balance'. The competitive atmosphere of the EDL could also have contributed to the decrease in fantasy through a focus on winning the EDL, versus rational decisions based on evidence. Participants may have been focused on how to best manipulate their allocations to receive the best score instead of imagining the plight of those in different sectors. A pragmatic suggestion from these results is to decide whether the goal of an engagement activity is to enhance the ability to imagine another's plight and thus ease the debate of potential trade-offs of decision making across a basin; or to maximize water managers' allocation skills for equal distribution of water. If the former, a competition-style engagement activity might act in opposition to allowing the development of more fantasy skills and thus, affective empathy, and instead reinforce existing values of opposing groups.

### 4.3. Empathic Concern and the EDL

Empathic concern is the ability to experience feelings of sympathy for the misfortunes of others, and as such represents an emotional component of empathy. In the EDL, however, no opportunity to share narratives that may invoke emotional responses were given except for a brief time in the final two rounds. During follow-up, participants did, however, report that they preferred being able to talk with others while making their decisions. The results of this study indicate that empathic concern decreased for all but government agency workers. This empathic concern construct has been reported to be higher in women in general, and to peak in middle-age for most people [51,52]. In this study, that trend was found, although no statistically significant changes were noted for these groups as a result of the EDL intervention. Significant decreases in empathic concern were apparent for men in the sample, potentially indicating decreased sympathy and care for other water stakeholders when participants are polarized and over-engaged in the competition, as found in other studies [1,3,8]. It is interesting that government agency workers experienced a small increase in empathic concern; however, other studies have noted that empathic concern may actually express one's awareness of what is an appropriate response in a particular context and not necessarily the mirroring of emotions displayed by others [53]. In this case, the government agency representatives may have reported an increase in empathic concern via their becoming aware of the appropriate response in a context that they are typically removed from, rather than actually feeling it. The idea of lack of autonomy of government workers thus partly explains the lack of increase in the other affective components. Others have also suggested that government

workers are exposed to increased pressures of social desirability on behalf of their agencies, and thus respond in ways to make agencies appear more positively attuned to situations where demonstrations of concern are expected [54]. In future iterations of the EDL, it would be appropriate to encourage autonomous responses for measuring increased individual empathy to avoid the social pressure on particular groups to maintain identity and consistency of messaging.

### 4.4. Gender and Empathy in the EDL

Overall, women scored themselves higher on the four empathic components to begin with, and in the post-test results, but the significant increase in perspective taking is an important finding. It indicates that, for women, the EDL has potential to act on cognitive components of empathy through increasing the ability to shift perspectives, or step outside the self [40]. Others have also demonstrated that gender differences in perspective taking exist, with women reporting markedly higher levels at all ages, and more perceived fairness in resource allocations when decision makers had more women in leadership roles [55,56]. That perspective taking increased (with a medium effect size) is an interesting finding, particularly because the EDL was designed as an individually competitive process. In qualitative findings, women reported deliberating about both strategies to win the EDL, and mutually beneficial actions while allocating water. Others have stressed the importance of including women in water allocations globally; in this case, the results support that assertion and if the goal is to increase water empathy, gender mainstreaming needs to occur [57].

The qualitative results also describe increases in the ability to understand the complicated context with multiple competing demands in which water allocation decision are made. We posited that results may also indicate a discursive process for perspective taking, especially for women who were more comprehensive in their open-ended answers; that is, claiming understanding of other's perspectives enables participants to justify their own arguments in a more socially persuasive way. With contested definitions of empathy ranging from affective concern, to deceitful emotional guises, the counter intuitive results call into question whether increasing empathy as measured using existing scales is accurate. Recent works develop the arguments in more detail for and against increasing empathy as a means for improving the world [58,59]. In any case, the results of the IRI perspective taking component highlights participants' learning about each other's needs, and, the reflective consideration of those needs during the EDL and afterwards. This result applies to women, urban residential water users, government agency representatives, and researchers, but not men, rural agricultural water users and those in the 'other' sector group.

### 4.5. EDL and Water Empathy

The answer to whether an EDL is an effective way of increasing empathy among water users in the Saskatchewan River Basin and elsewhere, and whether a scale such as the IRI is effective for measuring water empathy change is thus complicated. In this case, the EDL did result in an increase in perspective taking as measured through seven items on the IRI among participants (particularly women), but it decreased the affective components of empathy overall. The EDL was economically-based; that is, no qualitative implications for decisions were given, only a balance sheet of resources left to allocate for each round/drought/flood. Criticism of similar gaming approaches have suggested that consequences of serious gaming tools need to be multidimensional in nature to be effective for behaviour change [60,61]. Removing the personal components (i.e., narratives, media portrayals of people affected) sanitizes the human experience, thereby reducing the cues for which a more thorough empathy-building experience can occur. Other recent work has demonstrated that the personal components develop trust, which can lead to empathy [62], though the direction of the trust–empathy relationship is still under debate [63]. Expanding researcher and practitioner knowledge about the importance of many human dimensions of water security, including trust and empathy are important as decision makers need both technological advancements in water modeling and management, as well as advancements in socio-hydrological, and hydrosocial relationships to create the 'imaginative practitioners' called for by water experts [64].

*4.6. Implications in Engagement Activities for Transboundary Water Empathy Building*

When the two-stage development process of empathy is considered, the EDL provides some insight for researchers and practitioners looking to enhance water empathy in a basin or other transboundary context. First, this study found that the EDL did not increase affective resonance as evidenced by the affective components of the IRI. In short, the EDL did not invoke feelings in participants that might invoke changes in behavior during a competitive activity. This is not to say that decision laboratories do not work for enhancing empathy, but this study's finding is supported by results in other contexts involving screen-based simulations of real life. Autism studies, for example, demonstrate that interactive media tools can enhance recognition of mental states of others; however, additional, face-to-face methods are required to enhance empathic skills [65,66]. The EDL and other screen-based engagement activities as tools for water managers need further development in order to increase all components of empathy. Other research has demonstrated that screen-time interferes with the development of empathy, except in the case of using pro-social games-based media (video games where players must work together to achieve the goal) [67–69]. This builds the case for future group-based EDL work combined with in-person engagement exercises, as well as the need for advances in other in-person game-based methodologies for resource management stakeholders. Yes, computer usage (and apps) are credible for improving understanding of diverse stakeholder and sector needs for water management (i.e., the perspective taking part of empathy), but decision-laboratory scenarios should not be used in isolation from real interaction. Measurement of the effectiveness of those scenarios using scales like the IRI requires refinement. It may also be the case that water empathy is a construct that takes time to develop, and thus a one-time measurement over a one-time experience using an EDL is not adequate to assess its development.

Secondly, the results support the need for coming up with stakeholder engagement methods that promote the creation of a shared language for the conflicting values, beliefs, and needs of the different water sectors. The EDL did appear to enhance the cognitive component of empathy among some stakeholder groups, but, without the applied creation of a shared language, empathy gains did not translate to a quantifiable measure of affective resonance among stakeholders. The EDL had different communication settings. In the rounds where participants could communicate with each other, the allocation results ended up being most rewarding economically, resonating with others findings [70]. The participants in this study also indicated support for open communication in their qualitative responses. New empathy scales for use during engagement activities and/or empathy construct measurement tools in resource management contexts overall are needed, echoing the findings of others [71,72]. There may also be other confounding variables which can be measured using tools from psychology, such as trust, conformity, and groupthink; however, these constructs were beyond the scope of this particular study.

## 5. Conclusions

To avoid some of the pitfalls in resource management decision-making for shared resources such as transboundary basins, participants need support to learn to communicate with other water users identifying with different genders, riparian locations, and employment sectors. Overcoming habits built from past experience, enhancing social capital among groups and between individuals (of different genders, economic sectors and riparian regions) and gaining a full understanding of participants needs at the outset of resource management planning requires a variety of communication approaches. This study suggests that, to enhance water empathy in a holistic way (i.e., not just cognitively), experimental decision laboratory approaches are not sufficient. Without interpersonal connections separate from online or computer-based scenarios, learning to empathize with other stakeholders in transboundary riparian settings is unachievable across diverse stakeholder groups. Recommendations for improving water empathy include face-to-face interactions where real or perceived impacts of water management decision will be discussed from a variety of worldviews, and the resulting qualitative data can provide contextual details for how empathy is enacted.

We recognize the multiple limitations of this work, primarily in the use of a scale from the context of psychological development in a resource management sector. In the future, it would be useful to create and pilot a modified IRI or similar scale; to counterbalance and implement it between the rounds of the EDL instead of simply pre- and post-testing all participants for the entire EDL experience; and involving participants in the meta-analyses of the scale itself to test its validity in the water allocation context. It would also be useful to experiment with alternative IRI-like water empathy scales triangulated with a variety of qualitative methods such as interviews, focus groups, and asset mapping exercises. In any case, there is consensus that bringing stakeholders together to discuss their needs and challenges can improve overall understanding of water allocations in transboundary contexts, and hints towards what types of experiences contribute to enhancing water empathy. Though the participants were not allowed open communications until the last two rounds of the EDL competition, we realize that communication with others plays an important role in how allocation decisions are made. There is evidence that the stories people hear and carry with them, or the memories of past experiences with people in the room influence decision-making for resource management [73].

This study had other limitations. It was a pilot study in one post-industrialized transboundary basin. Participants had previous experience with the research team examining water security as a construct and learning about different basin sectors. Participants may have had previous experience working with each other. They may have also been fatigued due to ongoing engagement on a larger project. There was limited involvement of downstream users and Indigenous people in the experimental decision laboratory. The mobile EDL was a one-time venture. This study had a small sample size; thus, determining interaction effects was not possible; however, future work of this nature with larger sample sizes may be able to partly overcome this limitation through the use of counterbalancing. Confounding variables such as bias, experience with EDLs or water allocation systems, environmental values, experience working with Indigenous people, and others were unable to be examined at this point, but likely played a part in the results.

Other psychological scales for studying empathy exist and could provide other insights on whether aspects of empathy and prosocial behaviours not captured in the IRI are influenced by the EDL. The EDL itself is a tool still under development with potential for widespread use in various natural resource management sectors. Given the world's thirst, methods of increasing water empathy and measuring the progress towards a shared language and experience of water challenges in the changing world are worthy pursuits.

**Author Contributions:** Conceptualization, G.S. and L.B.; methodology, G.S., L.B., and R.B.; formal analysis, L.B., and K.P.C.; writing—original draft preparation, L.B.; writing—review and editing, all; visualization, K.P.C.; supervision, G.S.; project administration, L.B. and G.S.; funding acquisition, L.B. and G.S.

**Funding:** We'd like to acknowledge funding support from the Social Sciences and Humanities Research Council of Canada (Grant No. 430-213-000231).

**Acknowledgments:** We'd like to acknowledge the participants who undertook this pilot study—their insights were invaluable for our work. The authors would like to thank Dr. Kevin Moore (Lincoln, New Zealand) for insights on a early version of this manuscript. Alasdair Morrison assisted with citation management. The EDL would not have been possible without the expertise and equipment provided by the University of Saskatchewan Social Science Research Laboratory.

**Conflicts of Interest:** The authors declare no conflict of interest. The funders had no role in the design of the study; in the collection, analyses, or interpretation of data; in the writing of the manuscript, or in the decision to publish the results.

## Appendix A

Interpersonal Reactivity Index Survey for mobile EDL (FS Fantasy component, EC Empathic concern component, PT Perspective taking component, PD Personal distress component) ( '-' means phrased in the negative)

1. I daydream and fantasize, with some regularity, about things that might happen to me. (FS)
2. I often have tender, concerned feelings for people less fortunate than me. (EC)

3.　I sometimes find it difficult to see things from the "other guy's" point of view. (PT) (-)

4.　Sometimes I don't feel very sorry for other people when they are having problems. (EC) (-)

5.　I really get involved with the feelings of the characters in a novel. (FS)

6.　In emergency situations, I feel apprehensive and ill-at-ease. (PD)

7.　I am usually objective when I watched a movie or play, and I don't often get completely caught up in it (FS) (-)

8.　I try to look at everybody's side of a disagreement before I make a decision. (PT)

9.　When I see someone being taken advantage of, I feel kind of protective towards them. (EC)

10.　I sometimes feel helpless when I am in the middle of a very emotional situation. (PD)

11.　I sometimes try to understand my friends better by imagining how things look from their perspective (PT)

12.　Becoming extremely involved in a good book or movie is somewhat rare for me. (FS) (-)

13.　When I see someone get hurt, I tend to remain calm. (PD) (-)

14.　Other people's misfortunes do not usually disturb me a great deal. (EC) (-)

15.　If I'm sure I'm right about something, I don't waste much time listening to other people's arguments. (PT) (-)

16.　After seeing a play or movie, I have felt as though I were one of the characters. (FS)

17.　Being in a tense emotional situation scares me. (PD)

18.　When I see someone being treated unfairly, I sometimes don't feel very much pity for them. (EC) (-)

19.　I am usually pretty effective in dealing with emergencies. (PD) (-)

20.　I am often quite touched by things that I see happen. (EC)

21.　I believe that there are two sides to every question and try to look at them both. (PT)

22.　I would describe myself as a pretty soft-hearted person. (EC)

23.　When I watch a good movie, I can very easily put myself in the place of a leading character. (FS)

24.　I tend to lose control during emergencies. (PD)

25.　When I'm upset at someone, I usually try to "put myself in his shoes" for a while. (PT)

26.　When I am reading an interesting story or novel, I imagine how I would feel if the events in the story were happening to me. (FS)

27.　When I see someone who badly needs help in an emergency, I go to pieces. (PD)

28.　Before criticizing somebody, I try to imagine how I would feel if I were in their place. (PT).

## Appendix B

Follow up and open-ended survey questions related to empathy:

1.　Did completing the Decision-Space for Water Security change any of your answers to the survey that you completed before and after the experimental decision lab? (Yes/No)

2.　Tell us what changed (Open-ended).

3.　Please share any comments or feedback that you would like to provide (Open-ended).

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
