# Peer review of "Does Engagement Build Empathy for Shared Water Resources? Results from the Use of the Interpersonal Reactivity Index during a Mobile Water Allocation Experimental Decision Laboratory"

_water, doi:10.3390/w11061259_

Round 1

Reviewer 1 Report

Main issues: 

Small sample size (include power test / effect size)

N = 37 isn't a big sample size for a robust quantitative analysis, but the authors go even further to analyze on a sub-group level (gender, age and sector). Some of the groups barely meet the minimum condition required for analytical tests (n=5 for each group). It's recommended for the authors to include the power of the tests (effect sizes) as well in order to better support the numbers.

Use of parametric tests

For such a small sample size, parametric tests like pairwise t-tests and ANOVA aren't recommended, because they seldom meet the necessary underlying assumptions for such analyses. As it is, the authors do not provide any information on whether the assumptions are met (e.g. normality and homogeneity of variance). Non-parametric tests (e.g. Wilcoxon Signed Rank Test) might be a better alternative, as they are less stringent in terms of the required assumptions (but may be less sensitive to changes).

Computation of mean scores

The empathy scales are measured on 5-point scales, so why aren't the component scores computed back to 5-point scores as well? The authors added the scores together instead of averaging the scores, with the result that the range of scores becomes 29 (7 to 35). This makes the differences in their tables harder to interpret, and when you consider that the biggest difference is less than 5 on a 29-point scale (Affective components for Rural Agricultural Water Users), that's really not a big change in absolute terms (probably less than 1 point when converted back to a 5-point scale).

Contamination to experimental design

The authors mentioned that participants made "water allocation decisions under three different policy scenarios, through two different levels of water stress and under two different communication rules (can or cannot communicate with other users). There were six separate rounds of allocation" (p 4 of 15). Unfortunately, the description of the process in the end notes was blinded, and thus unavailable. Assuming the allocation was random, the order by which the conditions were administered, and the combination of conditions that different subjects were exposed to, could introduce noise to the design. For experiments, the aim should always be to have the treatment be the same, so that pre- and post-test measurements can be attributed to the treatment and not any confounding variables (such as how the treatment was administered).

Subject-to-subject interaction

Another major concern has to do with the treatment condition of whether subjects "can or cannot communicate with other users". An important assumption of statistical tests (whether parametric or non-parametric) is the independence of observations, i.e. data from one subject cannot influence the data from another subject. But the condition of allowing subjects to interact (which is also highlighted again in the qualitative results section, where subjects mentioned how talking to others affected them) throws this assumption out of the window, and makes the statistical analyses unreliable.

Sub-group analyses

The experimental design starts off with a pre- & post-test design, but then goes on to analyze subgroup comparisons. Both the small sample sizes for the subgroups and the potential differences in treatment across subgroups (which means the lack of random assignment which is required for an experiment) have been mentioned above. Results with ANOVA (and the issue with parametric tests) has also been mentioned, making the subgroup analyses not very robust.

Interaction effects

Another issue with the subgroup analyses is that they are tested independently, without considering any interaction effect. It's entirely possible that sector resultss can be dependent on age or gender, and vice versa. A better option might be to run with ANCOVA (e.g. running sector analysis while controlling for age and gender), although meeting the assumptions for using ANCOVA will continue to be an issue. 

Interpretation of results - testing effects

While the independent subgroup tests suggests possible differences in empathy because of EDL across subgroups, at least some of the differences might have been due to testing effect (i.e. results of the post-test change because of the pre-test), rather than any real differences. For instance, perspective taking might become easier the second time you take the empathy test, but intensity of emotions (for the affect-driven factors) might be reduced, because you are taking the same test for the second time. This actually coincides with some of the results the authors obtained, and can't be ruled out from the current design. The use of a more robust design, with some subjects being exposed to either only the pre-test or only the post-test, or some subjects not being exposed to the treatment (EDL) as a control group, will go some way to making the design more rigorous. Of course, this will then also require a larger sample size.

Author Response

We’d like to thank you for your valuable feedback on our paper. We have completed a large amount of new statistical analyses and have revised our paper to meet your suggestions. Below we indicate specifically how we have addressed your concerns. Thank you for making this a much more robust piece of work. We hope it satisfies your needs this time around.

Sincerely,

Senior Author, on behalf of the research team

Reviewer 1:

1) Small sample size does not given evidence of effect sizes:  We agreed with the reviewer we now provide effect sizes via converting effect sizes to d (i.e., eta-squared to Cohen’s d).

2) Use of non-parametric tests: For a robust quantitative analysis, we agree with the reviewer. Non-parametric tests are used. We have replaced the pairwise t-test results by the Wilcoxon Signed Rank Test, and the ANOVA results by the Kruskal-Wallis test. We provide changes in means, rank scores, significance and effect sizes where needed in the tables.

3) Computation of mean scores: We now compute the mean scores. We agree that this helps the reader establish directionality of changes.

4) Contamination to experimental design: We also recognize these limitations. We now discuss them in our results. In the conclusion, we provide more discussion of limitations and potential confounding variables.

5) Subject-to-subject interaction: We also recognize this limitation though we do better describe the protocol for the EDL in the methods section. We explore this as a limitation in the conclusions.

6) Interaction effects: We do believe the reviewer that this may be important in future iterations, however, with our small sample size, we are unable at this point to do more complicated models.

Reviewer 2: We thank you for your very uplifting and positive review.

Reviewer 3:

1) Participant table - we now include a new table 1 with participant details

2)Put the numbers along with percentages – we have used the counts of participants in all cases now and have avoided the use of percentages.

3) Analyses: We now do non-parameteric analysis. We have also provided more descriptive analysis in Section 3.3. We also include the employment sector for all those that contribute quotes to the analyses.

4) We follow the suggestion and a comparison is provided for the rural agricultural water users (who were mostly male).

5) limitations – we have expanded our limitations paragraph and included other ;limitations where appropriate in the results and discussion.

6) Restructuring has been done in the discussion with subheadings and with expanded explanations and reference to literature and relevance for water researchers and engagement practitioners.

Reviewer 2 Report

It is hypothesised that an increase in awareness and sensitivity of each other’s situation will lead to an increase in the equitable sharing of water resources and thus resource security. Drawing on a water empathy approach, the efficacy of psychological empathy scale was studied by involving water manager, stakeholders and rightholders in transboundary basin, as starting point to develop a tool to measure water empathy.  The participants used laptops to make water allocation decision in 4 experimental lab sessions under two different communication regimes:  can or cannot communicate with other users.  4 components of empathy were measured using an interpersonal reactivity index by applying pre-and post-surveys to participating water managers.  Both intersectoral (age and gender) and sector by sector empathy significantly changed. The results from the interpersonal reactivity index scale indicate that all participants empathy decreased as a result of the lab sessions. It was concluded that decision laboratories, especially those mediated by screens, interferes with development empathy. These findings, the authors argue, support the need for stakeholder methods the promote a shared language, beliefs and needs.

Understanding the relationship between resource security and empathy is extremely important field and it touches upon all domains within sustainable development. This paper is an important contribution. 

Author Response

(The authors gave the same response as above.)

Reviewer 3 Report

The study tries to identify levels of water empathy by using  an existing empathy scale from clinical psychology. With a pre- and post test study design wants to analyse if engagement increases water empathy.

The idea of the study is very timely and interesting. Results may give hints to participatory and collaborative processes in water resources management.

In general, the manuscript is written in a good manner.

Nevertheless I have some comments for improving the manuscript:

1) I miss a table specifying the participants of the study, that one can see how many of the participants came from city 1 and so forth and what were their properties (female, ...) Providing a table would also delete the description of the participants by using % (see Lines 136 ff) From my opinion the % is misleading giving the feeling that more participants were included in the study. The study only included 37 participants by describing in shares with 5% and 3% only meaning 1 to 2 people should be avoided.

2) Since the n of the study is rather small I have my doubts about the representativeness of the statistical analysis. I would rather go for a descriptive analysis which will also support your findings.

3) I liked the description of the section 3.3 the qualitative data and it gave good insight into the participants. Here I would be interested to highlight more the differences between the participants groups (.e.g. agricultural water users vs. femal urban...) this could maybe also support your findings and comparison in section 3.2.

4) In lines 303 ff you describe the limitations which the gaming character may has on the results. It is very important to highlight this limitation but I also see that you should in general highlight the limitations of the study in order to highlight its valuable contribution. This description should go a bit beyond the  limitations you mention in the conclusion. So, there is a need for a limitation section/paragraph.

5) In general your discussion focusses on the different dimension of empathy and only in the last part of the discussion come to the link of engagement and participatory processes. I would suggest to re-structure your discussion usinig sub-headings and then focussing on dimension of empathy, role of empathy  in participatory processes and the importance of acknowledging water empathy in transboundary water resources management. These headings are just a suggestion.

6) By restructuring the discussion part you are able to highlight more the relevance of your study. This may also then effect your conclusions which should be adapted to the new structure of the discussion.

Author Response

(The authors gave the same response as above.)

Round 2

Reviewer 1 Report

I thank the authors for the considerable work done in addressing my concerns, and most of them have indeed been addressed. I am especially satisfied with the effort to address my points on contamination. However, the following three important concerns are still unaddressed, :

1. Sub-group   analyses remain fragile - must be more robust, or claims considerably weakened.

2. Interaction   effects -again it remains entirely possible that sector results can be dependent on age or gender, and vice versa.  The paper must address this point before it can be published.

3. Interpretation      of results - testing effects

It remain possible that at least some of the differences among the subgroups       might have been due to testing effect (i.e. results of the post-test       change because of the pre-test), rather than any real differences. The experiment requires a more robust design.

I understand that the writers cannot rework the experiment. What must give way is then the claims that the writers make on the basis of the findings.

I urge the writers to rethink their claims, and see if they can address these three points.